# Learning to Extrapolate Knowledge:
# Transductive Few-shot Out-of-Graph Link Prediction

**Jinheon Baek[1], Dong Bok Lee[1], Sung Ju Hwang[1,2]**
KAIST[1], AITRICS[2], South Korea
{jinheon.baek, markhi, sjhwang82}@kaist.ac.kr

## Abstract

Many practical graph problems, such as knowledge graph construction and drug-drug interaction prediction, require to handle multi-relational graphs. However, handling real-world multi-relational graphs with Graph Neural Networks (GNNs) is often challenging due to their evolving nature, as new entities (nodes) can emerge over time. Moreover, newly emerged entities often have few links, which makes the learning even more difficult. Motivated by this challenge, we introduce a realistic problem of *few-shot out-of-graph link prediction*, where we not only predict the links between the seen and unseen nodes as in a conventional out-of-knowledge link prediction task but also between the unseen nodes, with only few edges per node. We tackle this problem with a novel transductive meta-learning framework which we refer to as *Graph Extrapolation Networks (GEN)*. GEN meta-learns both the node embedding network for inductive inference (seen-to-unseen) and the link prediction network for transductive inference (unseen-to-unseen). For transductive link prediction, we further propose a stochastic embedding layer to model uncertainty in the link prediction between unseen entities. We validate our model on multiple benchmark datasets for knowledge graph completion and drug-drug interaction prediction. The results show that our model significantly outperforms relevant baselines for out-of-graph link prediction tasks.[1]

## 1   Introduction

Graphs have a strong expressive power to represent structured data, as they can model data into a set of nodes (objects) and edges (relations). To exploit the graph-structured data which works on a non-Euclidean domain, several recent works propose graph-based neural architectures, referred to as *Graph Neural Networks (GNNs)* [8, 19]. While early works mostly deal with simple graphs with unlabeled edges, recently proposed relation-aware GNNs [34, 35] consider multi-relational graphs with labels and directions on the edges. These multi-relational graphs expand the application of GNNs to more real-world domains such as natural language understanding [23], modeling protein structure [12], drug-drug interaction prediction [62], retrosynthesis planning [37], to name a few.

Among multi-relational graphs, *Knowledge Graphs (KGs)*, which represent knowledge bases (KBs) such as Freebase [2] and WordNet [24], receive the most attention. They represent entities as nodes and relations among the entities as edges, in the form of a triplet: *(head entity, relation, tail entity)* (e.g. *(Louvre museum, is located in, Paris)*). Although knowledge graphs in general contain a huge amount of triplets, they are well known to be highly incomplete [25]. Therefore, automatically completing knowledge graphs, which is known as the *link prediction* task, is a practically important problem for KGs. Prior works tackle this problem, i.e. inferring missing triplets, by learning embeddings of entities and relations from existing triplets, and achieve impressive performances [4, 57, 9, 28, 27].

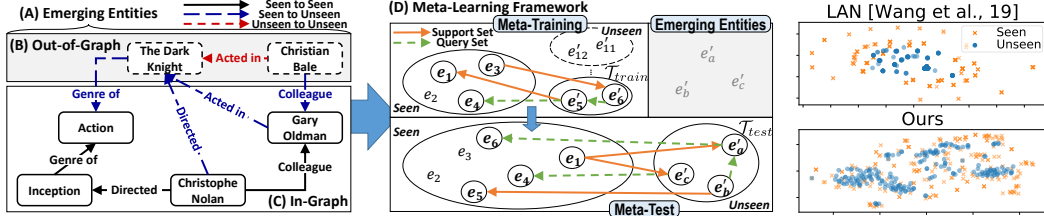

**Figure 1: Concept (Left):** An illustration of Out-of-Graph link prediction for emerging entities. Blue dotted arrows denote inferred relationships between seen and unseen entities, and red dotted arrows denote inferred relationships between unseen entities. **(Center):** An illustration of our meta-learning framework for the Out-of-Graph link prediction task. Orange arrows denote the support (training) set and green dotted arrows denote the query (test) set. **Visualization of the learned embeddings (Right):** Our transductive GEN embeds the unseen entities on the manifold of seen entities, while the baseline [48] embeds the unseen entities off the manifold.

Despite such success, the link prediction for KGs in real-world scenarios remains challenging for a couple of reasons. First, knowledge graphs dynamically evolve over time, rather than staying static. Shi and Weninger [36] report that around 200 new entities emerge every day. Predicting links on these emerging entities pose a new challenge, especially when predicting the links between emerging (unseen) entities them-

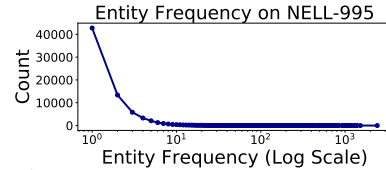

Figure 2: Entity frequency distribution.

selves. Moreover, real-world KGs generally exhibit long-tail distributions, where a large portion of the entities have only few triplets (See Figure 2). The embedding-based methods, however, usually assume that a sufficient number of associative triplets exist for training, and cannot embed unseen entities. Thus they are highly suboptimal for learning and inference on evolving real-world graphs.

Motivated by the limitations of existing approaches, we introduce a realistic problem of *Few-Shot Out-of-Graph* (OOG) link prediction for emerging entities. In this task, we not only predict the links between seen and unseen entities but also between the *unseen entities* themselves (Figure 1, left). To this end, we propose a novel meta-learning framework for OOG link prediction, which we refer to as *Graph Extrapolation Networks (GENs)* (Figure 1, center). GENs are meta-learned to extrapolate the knowledge from seen to unseen entities, and transfer knowledge from entities with many to few links.

Specifically, given embeddings of the seen entities for a multi-relational graph, we meta-train two GNNs to predict the links between seen-to-unseen, and unseen-to-unseen entities. The first GNN, *inductive GEN*, learns to embed the unseen entities that are not observed, and predicts the links between seen and unseen entities. The second GNN, *transductive GEN*, learns to predict the links not only between seen and unseen entities, but also between unseen entities themselves. This transductive inference is possible since our meta-learning framework can simulate the unseen entities during meta-training, while they are unobservable in conventional learning schemes. Also, since link prediction for unseen entities is inherently unreliable, which gets worse when few triplets are available for each entity, we learn the distribution of unseen representations for stochastic embedding to account for the uncertainty. Further, we apply a transfer learning strategy to model the long-tail distribution. These lead GEN to represent the unseen entities that are well aligned with the seen entities (Figure 1, right).

We validate GENs for their OOG link prediction performance on three knowledge graph completion datasets, namely FB15K-237 [2], NELL-995 [55], and WN18RR [9]. We also validate GENs for OOG drug-drug interaction prediction task on DeepDDI [31] and BIOSNAP-sub [63] datasets. The experimental results on five datasets show that our model significantly outperforms the baselines, even when they are retrained from scratch with unseen entities considered as seen entities. Further analysis of each component shows that both inductive and transductive layers of GEN help with the accurate link prediction for OOG entities. In sum, our main contributions are summarized as follows:

- We tackle a realistic problem setting of **few-shot out-of-graph link prediction**, aiming to perform link prediction not only between seen and unseen entities but also among unseen entities for multi-relational graphs that exhibit long-tail distributions, where each entity has only few triplets.
- To tackle this problem, we propose a **novel meta-learning framework**, Graph Extrapolation Network (GEN), which meta-learns the node embeddings for unseen entities, to obtain low error on link prediction for both seen-to-unseen (inductive) and unseen-to-unseen (transductive) cases.
- We validate GEN for few-shot out-of-graph link prediction tasks on **five benchmark datasets** for **knowledge graph completion** and **drug-drug interaction prediction**, on which it significantly outperforms relevant baselines, even when they are retrained with the unseen entities.

## 2 Related Work

**Graph Neural Network** Existing Graph Neural Networks (GNNs) encode the nodes by aggregating the features from the neighboring nodes, that use recurrent neural networks [15, 33], mean pooling with layer-wise propagation rules [19, 17], learnable attention-weighted combinations of the features [45, 29], to name a few. While most of the existing models work with simple undirected graphs, some recently proposed models tackle the multi-relational graphs for their practical importance. Directed-GCN [23] and Weighted-GCN [35] consider direction and relation types, respectively. Also, R-GCN [34] simultaneously considers direction and relation types. Similarly, MPNN [14] uses the edge-conditioned convolution to reflect the information on the edge types between nodes. Recently, Vashishth et al. [44] propose to jointly embed nodes and relations in a multi-relational graph. Since our GEN is a general framework for out-of-graph link prediction rather than a specific GNN architecture, it is compatible with any GNN implementations for multi-relational graphs.

**Meta Learning** Meta-learning, whose objective is to generalize over the distribution of tasks, is an essential approach for our few-shot out-of-graph link prediction framework, where we simulate the unseen nodes with a subset of training nodes. To mention a few, metric-based approaches [46, 38] learn a shared metric space to minimize the distance between correct and instance embeddings. On the other hand, gradient-based approaches [11, 30] learn shared parameters for initialization, to generalize over diverse tasks in a bi-level optimization framework. A few recent works consider meta-learning with GNNs, such as Satorras and Estrach [32] and Liu et al. [21] propose to meta-learn the GNNs for few-shot image classification, and Zhou et al. [61], Ding et al. [10] and Lan et al. [20] propose to meta-learn the GNNs for few-shot node classification. Further, Meta-Graph [5] proposes to construct graphs over the seen nodes, with only a small sample of known unlabeled edges.

**Multi-relational Graph** A popular application of multi-relation graphs is Knowledge Graph (KG) completion. Previous methods for this problem can be broadly classified as translational distance based [4, 50], semantic matching based [57, 43], convolutional neural network based [28, 9], and graph neural network based methods [34, 27]. While they require a large number of training instances to embed nodes and edges in a graph, many real-world graphs exhibit long-tail distributions. Few-shot relational learning methods tackle this issue by learning few relations of seen entities [56, 6, 58]. Nonetheless, the problem becomes more difficult as real-world graphs have an evolving nature with new emerging entities. Several models [54, 51] tackle this problem by utilizing extra information about the entities, such as their textual description. Furthermore, some recent methods [16, 48, 1] propose to handle unseen entities in an inductive manner, to generate embeddings for unseen entities without re-training the entire model from scratch. However, since they can not simulate the unseen entities in the training phase, there are some fundamental limitations on the generalization for handling actual unseen entities. On the other hand, our method entirely tackles both of seen-to-unseen and unseen-to-unseen link prediction, under the transductive meta-learning framework that simulates the unseen entities during training. Drug-Drug Interaction (DDI) prediction is another important real-world application of multi-relational graphs, where the problem is to predict interactions between drugs. Recently, Zitnik et al. [62] and Ma et al. [22] propose end-to-end GNNs to tackle this problem, which demonstrate comparatively better performances over non-GNN methods [47, 59, 60].

## 3 Few-Shot Out-of-Graph Link Prediction

Our goal is to perform link prediction for emerging entities of multi-relational graphs, in which a large portion of the entities have only few triplets associated with them. We begin with the definitions of the multi-relational graph and the link prediction task, which we formalize as follows:

**Definition 3.1 (Multi-relational Graph).** *Let $\mathcal{E}$ and $\mathcal{R}$ be two sets of entities and relations respectively. Then a link is defined as a triplet $(e_h, r, e_t)$, where $e_h, e_t \in \mathcal{E}$ are the head and the tail entity, and $r \in \mathcal{R}$ is a specific type of relation between the head and tail entities. A multi-relational graph $\mathcal{G}$ is represented as a collection of triplets. That is denoted as follows: $\mathcal{G} = \{(e_h, r, e_t)\} \subseteq \mathcal{E} \times \mathcal{R} \times \mathcal{E}$.*

**Definition 3.2 (Link Prediction).** *Link prediction refers to the task of predicting an unknown item of a triplet, when given two other items. We consider both of the entity prediction and relation prediction tasks. Entity prediction refers to the problem of predicting an unknown entity $e \subseteq \mathcal{E}$, given the entity and the relation: $(e_h, r, ?)$ or $(?, r, e_t)$. Relation prediction refers to the problem of predicting an unknown relation $r \subseteq \mathcal{R}$, given the head and tail entities: $(e_h, ?, e_t)$.*

**Link prediction for multi-relational graphs**   *Link prediction* is essentially the problem of assigning high scores to the true triplets, and therefore, many existing methods use score function $s(e_h, r, e_t)$ to measure the score of a given triplet, where the inputs depend on their respective embeddings (see Table 1). As a result, the objective of

Table 1: Score functions for multi-relational graphs, where $\oplus$ denotes concatenation.

| Model | Score Function | Domain |
|---|---|---|
| TransE [4] | $-\|e_h + r - e_t\|_2$ | Knowledge Graph |
| DistMult [57] | $\langle e_h, r, e_t \rangle$ | Knowledge Graph |
| Linear [14] | $r(e_h \oplus e_t)$ | Drug Interaction |

the link prediction is to find the representation of triplet elements and the function parameters in a parametric model case, which maximize the score of the true triplets. Which embedding methods to use depends on their specific application domains. However, existing works mostly tackle the link prediction between *seen* entities that already exist in the given multi-relational graph. In this work, we tackle a task of *few-shot Out-of-Graph (OOG)* link prediction formally defined as follows:

**Definition 3.3 (Few-Shot Out-of-Graph Link Prediction).** *Given a graph $\mathcal{G} \subseteq \mathcal{E} \times \mathcal{R} \times \mathcal{E}$, an unseen entity is an entity $e' \in \mathcal{E}'$, where $\mathcal{E} \cap \mathcal{E}' = \emptyset$. Then, out-of-graph link prediction is the problem of performing link prediction on $(e', r, ?)$, $(?, r, e')$, $(e', ?, \tilde{e})$, or $(\tilde{e}, ?, e')$, where $\tilde{e} \in (\mathcal{E} \cup \mathcal{E}')$. We further assume that each unseen entity $e'$ is associated with $K$ triplets: $|\{(e', r, \tilde{e})\text{ or }(\tilde{e}, r, e')\}| \leq K$ and $\tilde{e} \in (\mathcal{E} \cup \mathcal{E}')$, where $K$ is a small number (e.g. 1 or 3).*

While few existing works [16, 48, 1] tackle the entity prediction between seen and unseen entities, in real-world settings, unseen entities do not emerge one by one but may emerge simultaneously as a set, with only few triplets available for each entity. Thus, they are highly suboptimal in handling such real-world scenarios, such as few-shot out-of-graph link prediction which we tackle in this work.

# 4   Learning to Extrapolate Knowledge with Graph Extrapolation Networks

We now introduce *Graph Extrapolation Networks* (GENs) for the out-of-graph (OOG) link prediction task. Since most of the previous methods assume that every entity in the test set is *seen* during training, they cannot handle *emerging* entities, which are unobserved during training. While few existing works [16, 48, 1] train for seen-to-seen link prediction with the hope that the models generalize on seen-to-unseen cases, they are suboptimal in handling unseen entities. Therefore, we use the *meta-learning* framework to handle the OOG link prediction problem, whose goal is to train a model over a distribution of tasks such that the model generalizes well on unseen tasks. Figure 1 illustrates our learning framework. Basically, we meta-train GEN which performs both inductive and transductive inference on various simulated test sets of OOG entities, such that it *extrapolates* the knowledge of existing graphs to any unseen entities. We describe the framework in detail in next few paragraphs.

**Learning Objective**   Suppose that we are given a multi-relational graph $\mathcal{G} \subseteq \mathcal{E} \times \mathcal{R} \times \mathcal{E}$, which consists of seen entities $e \in \mathcal{E}$ and relations $r \in \mathcal{R}$. Then, we aim to represent the unseen entities $e' \in \mathcal{E}'$ over a distribution $p(\mathcal{E}')$, by extrapolating the knowledge on a given graph $\mathcal{G}$, to predict the links between seen and unseen entities: $(e, r, e')$ or $(e', r, e)$, or even between unseen entities themselves: $(e', r, e')$. Toward this goal, we have to maximize the score of a true triplet $s(e_h, r, e_t)$ that contains any unseen entities $e'$ to rank it higher than all the other false triplets, with embedding and score function parameters $\theta$ denoted as follows:

$$\max_{\theta} \mathbb{E}_{e' \sim p(\mathcal{E}')} \left[ s(e', r, \tilde{e}; \theta) \text{ or } s(\tilde{e}, r, e'; \theta) \right], \quad \text{where} \quad \tilde{e} \in (\mathcal{E} \cup \mathcal{E}') \text{ and } e' \in \mathcal{E}'. \quad (1)$$

While this is a seemingly impossible goal as it involves generalization to real unseen entities, we can tackle it with meta-learning by simulating unseen entities during training, which we describe next.

**Meta-Learning Framework**   While conventional learning frameworks can not handle unseen entities in the training phase, with meta-learning, we can formulate a set of tasks such that the model learns to generalize over unseen entities, which are simulated using seen entities. To formulate the OOG link prediction problem into a meta-learning problem, we first randomly split the entities in a given graph into the meta-training set for simulated unseen entities, and the meta-test set for real unseen entities. Then, we generate a task by sampling the set of simulated unseen entities during meta-training, for the learned model to generalize over actual unseen entities (See Figure 1, center).

Formally, each task $\mathcal{T}$ over a distribution $p(\mathcal{T})$ corresponds to a set of unseen entities $\mathcal{E}_{\mathcal{T}} \subset \mathcal{E}'$, with a predefined number of instances $|\mathcal{E}_{\mathcal{T}}| = N$. Then we divide the triplets associative with each entity $e'_i \in \mathcal{E}_{\mathcal{T}}$ into the support set $\mathcal{S}_i$ and the query set $\mathcal{Q}_i$: $\mathcal{T} = \bigcup_{i=1}^{N} \mathcal{S}_i \cup \mathcal{Q}_i$, where

**Algorithm 1** Meta-Learning of GEN

**Require:** Distribution over training tasks $p\left(\mathcal{T}_{train}\right)$
**Require:** Learning rate for meta-update $\alpha$
1: Initialize parameters $\Theta = \{\theta, \theta_\mu, \theta_\sigma\}$
2: **while** not done **do**
3:     Sample a task $\mathcal{T} \sim p\left(\mathcal{T}_{train}\right)$
4:     **for all** $e'_i \in \mathcal{T}$ **do**
5:         Sample support and query sets $\{\mathcal{S}_i, \mathcal{Q}_i\}$ correspond to $e'_i$
6:         Inductively generate using (3): $\phi_i = f_\theta\left(\mathcal{S}_i\right)$
7:     **end for**
8:     **for all** $e'_i \in \mathcal{T}$ **do**
9:         Transductively generate using (4): $\mu_i = g_{\theta_\mu}\left(\mathcal{S}_i, \phi\right)$ and $\sigma_i = g_{\theta_\sigma}\left(\mathcal{S}_i, \phi\right)$
10:        Sample $\phi'_i \sim \mathcal{N}\left(\mu_i, \mathrm{diag}\left(\sigma_i^2\right)\right)$
11:     **end for**
12:     Update $\Theta \leftarrow \Theta - \alpha \nabla_\Theta \sum_i \mathcal{L}\left(\mathcal{Q}_i; \phi'_i\right)$ using (6)
13: **end while**

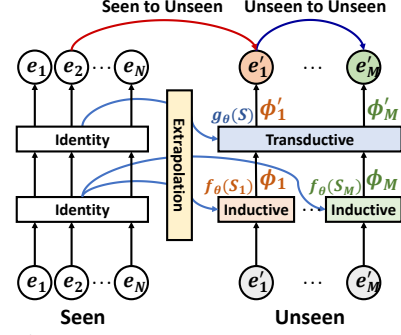

Figure 3: The overall framework of our model for each task. We extrapolate knowledge by using a support set $S$ with inductive and transductive learning, and then predict links with the output embedding $\phi'$.

$\mathcal{S}_i = \{(e'_i, r_j, \tilde{e}_j) \text{ or } (\tilde{e}_j, r_j, e'_i)\}_{j=1}^{K}$ and $\mathcal{Q}_i = \{(e'_i, r_j, \tilde{e}_j) \text{ or } (\tilde{e}_j, r_j, e'_i)\}_{j=K+1}^{M_i}; \tilde{e}_j \in (\mathcal{E} \cup \mathcal{E}')$. $K$ is the few-shot size, and $M_i$ is the number of triplets associated with each unseen entity $e'_i$. Our meta-objective is then learning to represent the unseen entities as $\phi$ using a support set $\mathcal{S}$ with a meta-function $f$, to maximize the triplet score on a query set $\mathcal{Q}$ with a score function $s$ as follows:

$$\max_\theta \mathbb{E}_{\mathcal{T} \sim p(\mathcal{T})} \left[ \frac{1}{N} \sum_{i=1}^{N} \frac{1}{|\mathcal{Q}_i|} \sum_{j=K+1}^{M_i} s(e'_i, r_j, \tilde{e}_j; \phi_i, \theta) \text{ or } s(\tilde{e}_j, r_j, e'_i; \phi_i, \theta) \right], \quad \phi_i = f_\theta(\mathcal{S}_i). \quad (2)$$

We refer to this specific setting as $K$-*shot out-of-graph (OOG) link prediction* throughout this paper. Once the model is trained with the meta-training tasks $\mathcal{T}_{train}$, we can apply it to unseen meta-test tasks $\mathcal{T}_{test}$, whose set of entities is disjoint from $\mathcal{T}_{train}$, as shown in the center of Figure 1.

**Graph Extrapolation Networks** In order to extrapolate knowledge of a given graph $\mathcal{G}$ to an unseen entity $e'_i$ through a support set $\mathcal{S}_i$, we propose a GNN-based meta-learner that outputs the representation of unseen entities. We formulate our meta-learner $f_\theta(\cdot)$ as follows (Figure 3-Inductive):

$$f_\theta\left(\mathcal{S}_i\right) = \frac{1}{K} \sum_{(r,e) \in n(\mathcal{S}_i)} \mathbf{W}_r C_{r,e}, \quad (3)$$

where $n(\cdot)$ is a set of neighboring entities and relations: $n\left(\mathcal{S}_i\right) = \{(r, e) \,|\, (e'_i, r, e) \text{ or } (e, r, e'_i) \in \mathcal{S}_i\}$. Further, $K$ is a size of $n(\mathcal{S}_i)$, $\mathbf{W}_r \in \mathbb{R}^{d \times 2d}$ is a relation-specific transformation matrix that is meta-learned, and $C_{r,e} \in \mathbb{R}^{2d}$ is a concatenation of feature representations of the relation-entity pair. Since GEN is essentially a framework for OOG link prediction, it is compatible with any GNNs.

**Transductive Meta-Learning of GENs** The previously described *inductive* GEN constructs the representation of each unseen entity $e'_i$ through a support set $\mathcal{S}_i$, and then performs link prediction on a query set $\mathcal{Q}_i$, independently. A major drawback of this inductive scheme is that it does not consider the relationships between *unseen* entities. However, to tackle unseen entities simultaneously as a set, one should consider not only the relationships between seen and unseen entities as with the inductive GEN, but also among unseen entities themselves. To tackle this issue, we extend the inductive GEN to further perform a transductive inference, which will allow knowledge to propagate between unseen entities (see supplementary materials for further discussions on inductive and transductive GENs).

More specifically, we add one more GEN layer $g_\theta(\cdot)$, which is similar to the inductive meta-learner $f_\theta(\cdot)$, to consider inter-relationships between unseen entities (Figure 3-Transductive):

$$g_\theta\left(\mathcal{S}_i, \phi\right) = \frac{1}{K} \sum_{(r,e) \in n(\mathcal{S}_i)} \mathbf{W}'_r C_{r,e} + \mathbf{W}_0 \phi_i, \quad (4)$$

where $\mathbf{W}_0 \in \mathbb{R}^{d \times d}$ is a weight matrix for the self-connection to consider the embedding $\phi_i$, which is updated by the previous inductive layer $f_\theta(\mathcal{S}_i)$. To leverage the knowledge of neighboring *unseen* entities, our transductive layer $g_\theta(\cdot)$ aggregates the representations across all the neighbors with a weight matrix $\mathbf{W}'_r \in \mathbb{R}^{d \times 2d}$, where neighbors can include the unseen entities with embeddings $\phi$, rather than treating them as noises or ignoring them as zero vectors like a previous inductive scheme.

**Stochastic Inference**   A naive transductive GEN generalizes to the unseen entities by simulating them with the seen entities during meta-training. However, due to the intrinsic unreliability of few-shot OOG link prediction with each entity having only few triplets, there could be high uncertainties on the representations of unseen entities. To model such uncertainties, we *stochastically* embed the unseen entities by learning the distribution over an unseen entity embedding $\phi_i'$. To this end, we first assume that the true posterior distribution has a following form: $p(\phi_i' \mid \mathcal{S}_i, \phi)$. Since computation of the true posterior distribution is intractable, we approximate the posterior using $q\left(\phi_i' \mid \mathcal{S}_i, \phi\right) = \mathcal{N}\left(\phi_i' \mid \mu_i, \mathrm{diag}\left(\sigma_i^2\right)\right)$, and then compute the mean and variance via two individual transductive GEN layers: $\mu_i = g_{\theta_\mu}\left(\mathcal{S}_i, \phi\right)$ and $\sigma_i = g_{\theta_\sigma}\left(\mathcal{S}_i, \phi\right)$, which modifies the GraphVAE [18] to our setting. The form to maximize the score function $s$ is then defined as follows:

$$s\left(e_h, r, e_t\right) = \frac{1}{L}\sum_{l=1}^{L} s\left(e_h, r, e_t; \phi'^{(l)}, \theta\right), \quad \phi'^{(l)} \sim q(\phi' \mid \mathcal{S}, \phi). \tag{5}$$

where we set the MC sample size to $L = 1$ during meta-training for computational efficiency. Also, we perform MC approximation with a sufficiently large sample size (e.g. $L = 10$) at meta-test. We let the approximate posterior same as the prior to make the consistent pipeline at training and test (see Sohn et al. [39]). We also model the source of uncertainty on the output embedding of an unseen entity from the transductive GEN layer via Monte Carlo dropout [13]. Our final *GEN* is then trained for both the inductive and transductive steps with stochastic inference, as described in Algorithm 1.

**Loss Function**   Each task $\mathcal{T}$ that corresponds to a set of unseen entities $\mathcal{E}_\mathcal{T} \subset \mathcal{E}'$ consists of a support set and a query set: $\mathcal{T} = \{\mathcal{S}, \mathcal{Q}\}$. During training, we represent the embeddings of unseen entities $e_i' \in \mathcal{E}_\mathcal{T}$ using the support set $\mathcal{S}$ with GENs. After that, at the test time, we use the true labeled query set $\mathcal{Q}_i$ to optimize our GENs. Since every query set contains only positive triplets, we perform negative sampling [4, 57] to update a meta-learner by allowing it to distinguish positive from negative triplets. Specifically, we replace the entity of each triplet in the query set: $\mathcal{Q}_i^- = \left\{(e_i', r, e^-) \text{ or } (e^-, r, e_i') \mid e^- \in \mathcal{E}\right\}$, where $e^-$ is the corrupted entity. In this way, $\mathcal{Q}_i^-$ holds negative samples for an unseen entity $e_i'$. We then use hinge loss to optimize our model as follows:

$$\mathcal{L}\left(\mathcal{Q}_i\right) = \sum_{(e_h, r, e_t) \in \mathcal{Q}_i} \sum_{(e_h, r, e_t)^- \in \mathcal{Q}_i^-} \max\left\{\gamma - s^+(e_h, r, e_t) + s^-(e_h, r, e_t)^-, 0\right\}, \tag{6}$$

where $\gamma > 0$ is a margin hyper-parameter, and $s$ is a specific score function in Table 1. $s^+$ and $s^-$ denote the scores of positive and negative triplets, respectively. Notably, for the drug-drug interaction predict task, we follow Ryu et al. [31] to optimize our model, where binary cross-entropy loss is calculated for each label, with a sigmoid output of the linear score function in Table 1.

**Meta-Learning for Long-Tail Tasks**   Since many real-world graphs follow the long-tail distributions (See Figure 2), it would be beneficial to transfer the knowledge from entities with many links to entities with few links. To this end, we follow a transfer learning scheme similar to Wang et al. [49]. Specifically, we start to learn the model with many shot cases, and then gradually decrease the number of shots to few shot cases in a logarithmic scale (see supplementary materials for details).

## 5   Experiment

We validate GENs on few-shot out-of-graph (OOG) link prediction for two different domains of multi-relational graphs: knowledge graph (KG) completion and drug-drug interaction (DDI) prediction.

### 5.1   Knowledge Graph Completion

**Datasets**   For knowledge graph completion datasets, we consider OOG *entity prediction*, whose goal is to predict the other entity given an unseen entity and a relation. **1) FB15k-237.** This dataset [42] consists of $310,116$ triplets from $14,541$ entities and $237$ relations, which is collected via crowdsourcing. **2) NELL-995.** This dataset [55] consists of $154,213$ triplets from $75,492$ entities and $200$ relations, which is collected by a lifelong learning system [26]. Since existing benchmark datasets do not target OOG link prediction, they assume that all entities given at the test time are seen during training. Therefore, we modify these two datasets such that the triplets used for link prediction at the test time contain at least one unseen entity (see supplemental materials for the detailed setup). **3) WN18RR.** This dataset [9] consists of $93,003$ triplets from $40,943$ entities and $11$ relations extracted from WordNet [24]. Particularly, this dataset includes the unseen entities in

Table 2: The results of 1- and 3-shot OOG link prediction on FB15k-237 and NELL-995. * means training a model within our meta-learning framework. Bold numbers denote the best results.

| | Model | FB15k-237 | | | | | | | | NELL-995 | | | | | | | |
|---|---|---|---|---|---|---|---|---|---|---|---|---|---|---|---|---|---|
| | | **MRR** | | **H@1** | | **H@3** | | **H@10** | | **MRR** | | **H@1** | | **H@3** | | **H@10** | |
| | | 1-S | 3-S | 1-S | 3-S | 1-S | 3-S | 1-S | 3-S | 1-S | 3-S | 1-S | 3-S | 1-S | 3-S | 1-S | 3-S |
| Seen to Seen | TransE [4] | .053 | .048 | .034 | .026 | .050 | .050 | .082 | .077 | .009 | .010 | .002 | .002 | .007 | .008 | .020 | .021 |
| | DistMult [57] | .017 | .014 | .010 | .009 | .019 | .014 | .029 | .022 | .017 | .016 | .009 | .008 | .017 | .017 | .029 | .028 |
| | R-GCN [34] | .008 | .006 | .004 | .003 | .007 | .005 | .011 | .010 | .004 | .004 | .001 | .001 | .003 | .003 | .007 | .006 |
| Seen to Seen (with Support Set) | TransE [4] | .071 | .120 | .023 | .057 | .086 | .137 | .159 | .238 | .071 | .118 | .037 | .061 | .079 | .132 | .129 | .223 |
| | DistMult [57] | .059 | .094 | .034 | .053 | .064 | .101 | .103 | .172 | .075 | .134 | .045 | .083 | .083 | .143 | .131 | .233 |
| | ComplEx [43] | .062 | .104 | .037 | .058 | .067 | .114 | .110 | .188 | .069 | .124 | .045 | .077 | .071 | .134 | .117 | .213 |
| | RotatE [40] | .063 | .115 | .039 | .069 | .071 | .131 | .105 | .200 | .054 | .112 | .028 | .060 | .064 | .131 | .104 | .209 |
| | R-GCN [34] | .099 | .140 | .056 | .082 | .104 | .154 | .181 | .255 | .112 | .199 | .074 | .141 | .119 | .219 | .184 | .307 |
| Seen to Unseen | MEAN [16] | .105 | .114 | .052 | .058 | .109 | .119 | .207 | .217 | .158 | .180 | .107 | .124 | .173 | .189 | .263 | .296 |
| | LAN [48] | .112 | .112 | .057 | .055 | .118 | .119 | .214 | .218 | .159 | .172 | .111 | .116 | .172 | .181 | .255 | .286 |
| | GMatching [56] | .093 | .105 | .061 | .061 | .100 | .112 | .146 | .183 | .060 | .079 | .051 | .059 | .063 | .097 | .076 | .106 |
| | MetaR [6] | .076 | .084 | .043 | .041 | .084 | .093 | .133 | .164 | .092 | .096 | .059 | .060 | .107 | .115 | .154 | .166 |
| | FSRL [58] | .097 | .090 | .065 | .058 | .104 | .096 | .156 | .150 | .067 | .085 | .054 | .064 | .068 | .095 | .091 | .126 |
| Ours | GMatching* [56] | .224 | .238 | .157 | .168 | .249 | .263 | .352 | .372 | .120 | .139 | .074 | .092 | .136 | .151 | .215 | .235 |
| | MetaR* [6] | .294 | .316 | .223 | .235 | .318 | .341 | .441 | .492 | .177 | .213 | .104 | .145 | .217 | .247 | .315 | .352 |
| | FSRL* [58] | .255 | .259 | .187 | .186 | .279 | .281 | .391 | .404 | .130 | .161 | .075 | .106 | .145 | .181 | .253 | .275 |
| | I-GEN | .348 | .367 | .270 | .281 | .382 | .407 | .504 | .537 | .278 | .285 | .206 | .214 | .313 | .322 | .416 | .426 |
| | T-GEN | **.367** | **.382** | **.282** | **.289** | **.410** | **.430** | **.530** | **.565** | **.282** | **.291** | **.209** | **.217** | **.320** | **.333** | **.421** | **.433** |

validation and test sets, which overlaps with the 16 triplets to evaluate on a query set during meta-test. Therefore, we compare models only on these 16 triplets. Detailed descriptions of each dataset are reported in the supplemental materials.

**Baselines and our models** **1) TransE, 2) RotatE.** Translation distance based embedding methods for multi-relational graphs [4, 40]. **3) DistMult, 4) ComplEx.** Semantic matching based embedding methods [57, 43]. **5) R-GCN.** GNN-based method for modeling multi-relational data [34]. **6) MEAN, 7) LAN.** GNN-based methods for a out-of-knowledge base task, which tackle unseen entities without meta-learning [16, 48]. **8) GMatching, 9) MetaR, 10) FSRL.** Link prediction methods for unseen relations of *seen entities*, which we further train with our meta-learning framework [56, 6, 58]. **11) I-GEN.** An inductive version of our GEN which is meta-learned to embed an unseen entity. **12) T-GEN.** A transductive version of GEN, with additional stochastic transductive GNN layers to predict the link between unseen entities. We report detailed descriptions in the supplementary file.

**Implementation Details** **1) Seen to Seen.** This scheme only trains seen-to-seen triplets from a meta-training set, without including unseen entities on a meta-test set. **2) Seen to Seen (with Support Set).** Following Xiong et al. [56], this scheme trains seen-to-seen link prediction baselines including support triplets of meta-validation and meta-test sets with unseen entities, since baselines are unable to solve the completely unseen entities at the test time. **3) Seen to Unseen.** This scheme tackles the link prediction for unseen entities without meta-learning [16, 48], or link prediction for unseen relations of seen entities with meta-learning [56, 6, 58]. **4) Ours.** Our meta-learning framework trains models only with a meta-training set, where we generate OOG entities using the episodic training [38]. For both I-GEN and T-GEN, we use **DistMult** for the initial embeddings of entities and relations, and the score function. We report detailed experimental setups in the supplementary file.

**Evaluation Metrics** For evaluation, we use the ranking procedure by Bordes et al. [3]. For a triplet with an *unseen* head entity, we replace its corresponding tail entity with candidate entities from the dictionary to construct corrupted triplets. Then, we rank all the triplets, including the correct and corrupted ones by a scoring measure, to obtain a rank of the correct triplet. We provide the results using mean reciprocal rank (**MRR**) and Hits at $n$ (**H@n**). Moreover, as done in previous works [4, 34], we measure the ranks in a filtered setting, where we do not consider triplets that appeared in either training, validation, or test sets. Finally, for a fair evaluation [41], we validate our models on different evaluation protocols, across which performances of our models remain consistent.

**Main Results** Table 2 shows that our I- and T-GEN outperform all baselines by impressive margins. Baseline models work poorly on emerging entities, even when they have seen the entities during training (with Support Set in Table 2). However, in our meta-learning framework, our GENs show superior performances over the baselines, even with a 1-shot setting. Moreover, while unseen relation prediction baselines achieve extremely low performances compared to our GENs, we train baselines in our meta-learning framework and obtain significantly improved results. However, their performances are still substantially lower than GENs, which shows that GEN's dedicated embedding layers for seen-to-unseen and unseen-to-unseen link prediction are more effective for OOG link prediction.

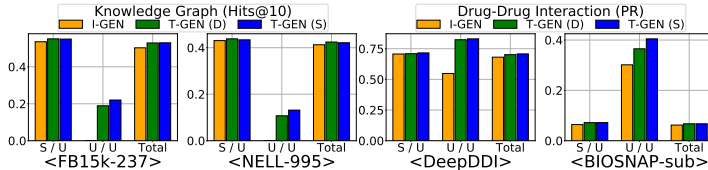
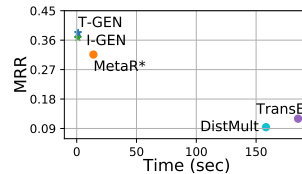

Figure 4: The results of seen to unseen (S/U), unseen to unseen (U/U) and total link prediction of I- and T-GEN with deterministic (D) and stochastic (S) modeling on KG completion and DDI prediction tasks.

Figure 5: 3-shot OOG link prediction results, reported with MRR over training time.

**Analysis on Seen to Unseen and Unseen to Unseen**   We observe that T-GEN outperforms I-GEN on both datasets by all evaluation metrics in Table 2. To see where the performance improvement comes from, we further examine the link prediction results for seen-to-unseen and unseen-to-unseen cases. Figure 4 shows that T-GEN obtains significant performance gain on the unseen-to-unseen link prediction problems, whereas I-GEN mostly cannot handle the unseen-to-unseen case as it does not consider the relationships between unseen nodes. Further, T-GEN with stochastic inference obtains even higher unseen-to-unseen link prediction performances, over deterministic T-GEN, which shows that modeling uncertainty in the latent embedding space of the unseen entities is effective.

**Efficiency of Meta-Learning**   To demonstrate the efficiency of our meta-learning framework that embeds unseen entities without additional re-training, we compare GENs against models trained from scratch including unseen entities, for 3-shot OOG link prediction on FB15k-237. Figure 5 shows that GENs largely outperform baselines with a fraction of time required to embed unseen entities. Also, MetaR trained in our meta-learning framework is slower since it uses additional gradient information. This shows that GENs are efficient and generalize well to unseen entities with effective GNN layers.

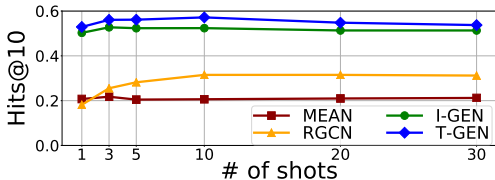

Figure 6: Diverse shots link prediction results with baselines and GENs on KG completion tasks.

Table 3: Cross-shot learning results of T-GEN on KG completion tasks, by varying training and test shots.

| Test | (Training) 1-Shot | | | (Training) 3-Shot | | |
|---|---|---|---|---|---|---|
| | MRR | H@1 | H@10 | MRR | H@1 | H@10 |
| 1-S | .367 | .282 | .530 | .346 | .262 | .507 |
| 3-S | **.377** | **.288** | .556 | **.382** | **.289** | .565 |
| 5-S | .362 | .266 | **.562** | .370 | .269 | **.570** |
| R-S | .375 | .287 | .548 | .373 | .282 | .547 |

**Robustness on Many Shots**   While we mostly target a long-tail graph with the majority of entities having few links, our method works well on many-shot cases as well (Figure 6), on which GENs still largely outperform the baselines, even though R-GCN sees the unseen entities during training.

**Robustness on Varying Shots**   We experiment our GEN with varying the number of shots by considering 1-, 3-, 5-, and random-shot (R-S: between 1 and 5) during meta-training and meta-test. Table 3 shows that differences in the number of shots used for training and test does not significantly affect the performance, which demonstrates the robustness of GENs on varying shots at test time. Moreover, our model trained on a 1-shot setting obtains even better performance on a 3-shot setting.

Table 4: 1-shot OOG link prediction results on WN18RR for unseen entities.

| Model | WN18RR | | |
|---|---|---|---|
| | MRR | H@1 | H@10 |
| DistMult [57] | .000 | .000 | .000 |
| TransE [4] | .011 | .000 | .031 |
| MetaR* [6] | .066 | .063 | .063 |
| I-GEN | **.125** | **.125** | **.125** |

Table 5: Ablation study of T-GEN on FB15k-237. **SI** means whether to apply stochastic inference.

| Model | SI | Seen to Unseen | | Unseen to Unseen | |
|---|---|---|---|---|---|
| | | MRR | H@3 | MRR | H@3 |
| T-GEN | O | .379 | .424 | **.185** | **.187** |
| w/o transfer strategy | O | .374 | .414 | .183 | .175 |
| w/o pretrain | O | .361 | .400 | .168 | .164 |
| w/o stochastic inference | X | **.384** | **.425** | .153 | .158 |
| w/o transductive scheme | X | .366 | .403 | .000 | .000 |

**Results on Unseen Entities for WN18RR**   As previously mentioned, WN18RR dataset includes a small number of unseen entities in the test set. Therefore, we validate GEN only against test triplets that contain unseen entities in the test set. Table 4 shows that our GEN can improve the performance of out-of-graph link prediction even on this benchmark dataset.

**Ablation Study**   We conduct an ablation study of the T-GEN on seen-to-unseen and unseen-to-unseen cases. Table 5 shows that using stochastic inference on the transductive layer helps significantly

improve the unseen-to-unseen link prediction performance. Moreover, the meta-learning strategy of learning on entities with many links and then progressing to entities with few links performs well. Finally, we observe that using pre-trained embedding of a seen graph leads to better performance.

**Qualitative Results**  We visualize the output representations of unseen entities with seen entities. Figure 1 (Right) shows that the embeddings of unseen entities are well aligned with the seen entities. Regarding concrete examples of the link prediction on NELL-995, please see supplementary file.

## 5.2 Drug-Drug Interaction

**Datasets**  We further validate our GENs on the OOG *relation prediction* task using two public Drug-Drug Interaction (DDI) datasets. **1) DeepDDI.** This dataset [31] consists of 1,861 drugs (entities) and 222,127 drug-drug pairs (triplets) from DrugBank [52], where 113 different relation types are used as labels. **2) BIOSNAP-sub.** This dataset [63, 22] consists of 645 drugs (entities) and 46,221 drug-drug pairs (triplets), where 200 different relation types are used as labels. Similar to the experiments on OOG knowledge graph completion tasks, we modify drug-drug interaction datasets for the OOG link prediction task. We report the detailed setup in supplemental materials.

**Baselines and our models**  **1) MLP.** Feed-forward neural network used in the DDI task [31]. **2) MPNN.** Graph Neural Network that uses edge-conditioned convolution operations [14]. **3) R-GCN.** The same model used in the entity prediction on the KG completion task [34]. **4) I-GEN.** Inductive GEN, which only uses a feature representation of an entity $e_k$, instead of a relation-entity pair $(r_k, e_k)$. This is because the relation is the prediction target for the DDI tasks. **5) T-GEN.** Transductive GEN with an additional transductive stochastic layer for unseen-to-unseen relation prediction.

**Implementation Details and Evaluation Metrics**  For both I-GEN and T-GEN, we use **MPNN** for the initial embeddings of entities with a linear score function in Table 1. To train baselines, we use the Seen to Seen (with Support Set) scheme as in the KG completion task, where support triplets of meta-validation and meta-test sets are included during training. We report detailed experimental settings in a supplementary file. For evaluation, we use the area under the receiver operating characteristic curve (**ROC**), the area under the precision-recall curve (**PR**), and the classification accuracy (**Acc**).

**Main Results**  Table 6 shows the Drug-Drug Interaction (DDI) prediction performances of the baselines and GENs. Note that the performances on BIOSNAP-sub are comparatively lower in comparison to DeepDDI, due to the use of the preprocessed input features, as suggested by Ryu et al. [31]. Similar to the KG completion tasks, both I- and T-GEN outperform all baselines by impressive margins in all evaluation metrics. These results demonstrate that our GENs can be easily extended to OOG link prediction for other real-world applications of multi-relational graphs.

Table 6: The results of 3-shot *relation* prediction on DeepDDI and BIOSNAP-sub.

| Model | DeepDDI | | | BIOSNAP-sub | | |
|---|---|---|---|---|---|---|
| | ROC | PR | Acc | ROC | PR | Acc |
| MLP | .928 | .476 | .528 | .597 | .034 | .049 |
| MPNN [14] | .939 | .478 | .681 | .597 | .026 | .067 |
| R-GCN [34] | .928 | .397 | .640 | .594 | .041 | .051 |
| I-GEN | .946 | .681 | .807 | .608 | .062 | .073 |
| T-GEN | **.954** | **.708** | **.815** | **.625** | **.067** | **.089** |

**Analysis on Seen to Unseen and Unseen to Unseen**  We also compare the link prediction performance for both seen-to-unseen and unseen-to-unseen cases on two DDI datasets. The rightmost two columns of Figure 4 show that T-GEN obtains superior performance over I-GEN on unseen-to-unseen link prediction cases, especially when utilizing stochastic modeling schemes.

## 6 Conclusion

We formally defined a realistic problem of the *few-shot out-of-graph (OOG)* link prediction task, which considers link prediction not only between seen to unseen (or emerging) entities but also between unseen entities for multi-relational graphs, where each entity comes with only few associative triplets to train. To this end, we proposed a novel meta-learning framework for OOG link prediction, which we refer to as *Graph Extrapolation Network (GEN)*. Under the defined $K$-shot learning setting, GENs learn to extrapolate the knowledge of a given graph to unseen entities, with a stochastic transductive layer to further propagate the knowledge between the unseen entities and to model uncertainty in the link prediction. We validated the OOG link prediction performance of GENs on five benchmark datasets, on which proposed model largely outperformed the relevant baselines.

## Broader Impact

Constructing knowledge bases that accurately reflect up-to-date knowledge about the entities and the links between them is crucial for its application in real-world scenarios. However, conventional link prediction methods for knowledge base systems mostly consider static knowledge graph that does not change over time. Yet, as new entities emerge every day [36] (e.g. COVID-19), the ability to dynamically incorporate them into the existing knowledge graph is becoming a significantly important problem, which we mainly tackle in this paper.

As a specific example of our approach, the novel coronavirus, COVID-19, is threatening our lives around the globe. To eradicate the novel coronavirus, we may want to best utilize the accumulated knowledge about existing coronavirus variants [53, 7] by identifying the links between the seen (SARS and MERS) and unseen entities (COVID-19), or the links between unseen entities that have newly emerged (COVID-19 and novel vaccine understudy). The following are more use cases of our proposed out-of-graph link prediction system:

- The proposed meta-learning based few-shot out-of-graph link prediction method can infer and inform the relationship between the entities that describe past coronavirus outbreaks and the current COVID-19 situation.
- Our transductive inference, with stochastic transductive GENs, can lead to finding the relationships among novel entities regarding COVID-19 that rapidly emerge over time, which may allow us to discover meaningful links among them.
- Regarding drug-drug interaction prediction, our method can be further utilized to analyze the side-effects of simultaneously taking novel antiviral drugs for COVID-19 and existing drugs, before the clinical trials.

While we describe the impact of our method on a specific, but significantly important topic, our method can be broadly applied to any real-world applications that require to predict the links which involve unseen entities. While our method obtains significantly better performance over existing methods on out-of-graph link prediction, its prediction performance is yet far from perfect. Thus, the model should be used as a candidate selection tool (Hits@N) when inferring critical information (e.g. drug-drug interaction prediction for COVID-19), and more efforts should be made to develop a reliable system.

## Acknowledgments and Disclosure of Funding

We thank the anonymous reviewers for their constructive comments and suggestions. This work was supported by Samsung Advanced Institute of Technology (SAIT), Seoul R&BD Program (IC190048), National Research Foundation of Korea (NRF) grant funded by the Korea government (MSIT) (NRF-2018R1A5A1059921), and Institute of Information & communications Technology Planning & Evaluation (IITP) grant funded by the Korea government (MSIT) (No.2016-0-00563, Research on Adaptive Machine Learning Technology Development for Intelligent Autonomous Digital Companion, and No.2019-0-00075, Artificial Intelligence Graduate School Program (KAIST)).

## Footnotes

[1]Code is available at https://github.com/JinheonBaek/GEN

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
