[Supplementary Material]

# [Supplementary File]
# Learning to Extrapolate Knowledge: Transductive Few-shot Out-of-Graph Link Prediction

**Organization**   The supplementary file is organized as follows. In Section A, we describe the detailed experimental setup which covers datasets, models, and implementation details. Also, we provide implementation details and additional experimental results on the meta-learning strategy for long-tail tasks in Section B. Section C provides additional experimental results with our graph extrapolation networks (GENs) under various settings, and Section D shows concrete examples of Out-of-Graph (OOG) link prediction on the knowledge graph completion dataset. Finally, we discuss the inductive and transductive schemes of our meta-learning framework in Section E.

## A   Experimental Setup

### A.1   Datasets

Since existing benchmark datasets assume that all entities given at the test time are seen during training, we modify the datasets to formulate the Out-of-Graph (OOG) link prediction task, where completely unseen entities appear at the test time. Dataset modification processes are as follows:

- First, we randomly sample the unseen entities, which have a relatively small amount of triplets on each dataset. We then divide the sampled unseen entities into meta-training/validation/test sets.
- Second, we select the triplets which are used for constructing an In-Graph, where the head and tail entities of every triplet in the In-Graph consist of only seen entities, not any unseen entity.
- Finally, we match the unseen entities in the meta-sets with their triplets. Each triplet in meta-sets contains at least one unseen entity. Also, every triplet in meta-sets is not included in the In-Graph.

Figure 1: Distribution for entity occurrences on FB15k-237, NELL-995, DeepDDI, and BIOSNAP-sub datasets.

**1) FB15k-237.**   This dataset [18] consists of 14,541 entities and 237 relations, which is collected from crowdsourcing and used for the knowledge graph completion task. We randomly sample the 5,000 entities from 10,938 entities, which have associated triplets between 10 and 100. Also, we split the entities such that we have 2,500/1,000/1,500 unseen (Out-of-Graph) entities and 72,065/6,246/9,867 associated triplets containing unseen entities for meta-training/validation/test. The remaining triplets that do not hold an unseen entity are used for constructing an In-Graph. As shown in the Figure 1, this dataset follows a highly long-tailed distribution.

**2) NELL-995.**   This dataset [24] consists of 75,492 entities and 200 relations, which is collected by a lifelong learning system [13] and used for the knowledge graph completion task. We randomly sample the 3,000 entities from 5,694 entities, which have associated triplets between 7 and 100. Also, we split the entities such that we have 1,500/600/900 unseen (Out-of-Graph) entities and 22,345/3,676/5,852 associated triplets containing unseen entities for meta-training/validation/test.

The remaining triplets that do not hold an unseen entity are used for constructing an In-Graph. As shown in the Figure 1, this dataset follows a highly long-tailed distribution.

**3) WN18RR.** This dataset [3] consists of 93,003 triplets from 40,943 entities and 11 relations, which is collected from WordNet [12] and used for the knowledge graph completion task. Particularly, this dataset essentially contains 198 unseen entities over 210 triplets on the validation set and 209 unseen entities over 210 triplets on the test set.

Note that, to construct a support set for training and a query set for test in our meta-learning framework, we need at least two triplets for each unseen entity. Therefore, even in the WN18RR that contains an appropriate number of unseen entities, the amount of triplets to evaluate on a query set is too small (only 16 triplets to test, which is 0.02 % compared to the number of all triplets), in which we consider validation and test sets together since test set only has 3 triplets to test. In other words, most unseen entities on WN18RR have only one triplet, which reflects the long-tail distribution of real-world graphs for emerging entities. Thus, we compare models only on these 16 triplets during meta-test.

To use the meta-learning framework from the conventional learning scheme, we randomly sample the 3,000 unseen entities from 4,478 entities, which have associated triplets between 8 and 100. After that, we use the sampled 3,000 unseen entities for meta-training which has 36,166 overlapped triplets. The remaining triplets that do not hold an unseen entity are used for constructing an In-Graph.

**4) DeepDDI.** This dataset [15] consists of 1,861 entities and 113 relations, which is collected from the DrugBank database [23] and used for the drug-drug interaction prediction task. We randomly sample the 500 entities from 1,039 entities, which have associated triplets between 7 and 300. Also, we split the entities such that we have 250/100/150 unseen (Out-of-Graph) entities and 27,726/1,171/2,160 associated triplets containing unseen entities for meta-training/validation/test. The remaining triplets that do not hold an unseen entity are used for constructing an In-Graph.

**5) BIOSNAP-sub.** This dataset [10] consists of 637 entities and 200 relations, which is collected from the BIOSNAP [28], further modified by Ma et al. [10] for efficiency and used for the drug-drug interaction prediction task. We randomly sample the 150 entities from 507 entities, which have associated triplets between 7 and 300. Also, we split the entities such that we have 75/30/45 unseen (Out-of-Graph) entities and 7,140/333/643 associated triplets containing unseen entities for meta-training/validation/test. The remaining triplets that do not hold an unseen entity are used for constructing an In-Graph.

### A.2  Baselines and Our Models

**Knowledge Graph Completion**  We describe the baseline models and our graph extrapolation networks for few-shot out-of-graph *entity prediction* on the knowledge graph (KG) completion task.

**1) TransE.** This is a translation embedding model for relational data by Bordes et al. [1]. It represents both entities and relations as vectors in the same space, where the relation in a triplet is used as a translation operation between the head and the tail entity.

**2) RotatE.** This model represents entities as complex vectors and relations as rotations in a complex vector space [17], which extends TransE with a complex operation.

**3) DistMult.** This model represents the relationship between the head and the tail entity in a bi-linear formulation, which can capture pairwise interaction between entities [26].

**4) ComplEx.** This model extends the DistMult by introducing embeddings on a complex space to consider asymmetric relations, where scores are measured based on the order of the entities [19].

**5) R-GCN.** This is a GNN-based method for modeling relational data, which extends the graph convolutional network to consider multi-relational structures, by Schlichtkrull et al. [16].

**6) MEAN.** This model computes the embedding of entities by GNN-based neighboring aggregation scheme, where they only train for seen-to-seen link prediction, with the hope that the model generalizes on seen-to-unseen cases, without meta-learning [6].

**7) LAN.** This model extends the MEAN [6] to consider relations with neighboring information by utilizing attention mechanisms, without meta-learning [21].

**8) GMatching.** This model tackles the link prediction on unseen relations of *seen entities* by searching for the closest entity pair with meta-learning [25]. We further extend it in our meta-learning framework such that it can handle unseen entities.

**9) MetaR.** This model tackles the link prediction on unseen relations of *seen entities* by generating the embeddings of unseen relations with gradient information over the meta-learning framework [2]. We further extend it in our meta-learning framework such that it can handle unseen entities.

**10) FSRL.** This model extends the GMatching [25] to tackle the link prediction on unseen relations of *seen entities* by utilizing attention mechanisms with meta-learning [27]. We further extend it in our meta-learning framework such that it can handle unseen entities.

**11) I-GEN.** This is an inductive version of our Graph Extrapolation Network (GEN), that is meta-learned to embed an unseen entity to infer hidden links between seen and unseen entities.

**12) T-GEN.** This is a transductive version of GEN, with additional stochastic transductive GNN layers on top of the I-GEN, that is meta-learned to predict the links between unseen entities as well as between seen and unseen entities.

**Drug-Drug Interaction** We describe the baseline models and our graph extrapolation networks for few-shot out-of-graph *relation prediction* on the drug-drug interaction (DDI) task.

**1) MLP.** This is a feed-forward neural network model used for DeepDDI [15] dataset. It classifies the relation of two drugs using their pairwise features.

**2) MPNN.** This is a GNN-based model that uses features for relation types with edge-conditioned convolution operations [5].

**3) R-GCN.** This is the same model used in the entity prediction on knowledge graph completion tasks [16], applied to drug-drug interaction tasks.

**4) I-GEN.** This is an inductive GEN, which only uses the feature representation of the entity $e_k$ when aggregating neighboring information, instead of using the concatenated representation of the relation-entity pair $(r_k, e_k)$ like KG completion tasks. This is because the relation is the prediction target for the DDI tasks.

**5) T-GEN.** This is a transductive version of GEN, with additional transductive stochastic layers on top of the I-GEN, for unseen-to-unseen relation prediction as well as seen-to-unseen prediction.

### A.3 Implementation Details

For every dataset, we set the embedding dimension for both entity and relation as 100. Also, we set the embedding of unseen entities as the zero vector. Furthermore, since we consider a highly multi-relational graph, we use the basis decomposition on weight matrices $\mathbf{W}_r$ and $\mathbf{W}'_r$ to prevent the excessive increase in the model size, which is proposed in Schlichtkrull et al. [16]: $\mathbf{W}_r = \sum_{b=1}^{B} a_{r_b} \mathbf{V}_b$, where $B$ is a number of basis, $a_{r_b}$ is a coefficient of each relation $r \in \mathcal{R}$, and $\mathbf{V}_b \in \mathbb{R}^{d \times 2d}$ is a shared representation of various relations. For all experiments, we use PyTorch [14] and PyTorch geometric [4] frameworks, and train on a single Titan XP or a single GeForce RTX 2080 Ti GPU. We optimize the proposed GENs using Adam [8].

**Knowledge Graph Completion** For both I-GEN and T-GEN, we search for the learning rate $\alpha$ in the range of $\{3 \times 10^{-4}, 1 \times 10^{-3}, 3 \times 10^{-3}\}$, margin $\gamma$ in the range of $\{0.25, 0.5, 1\}$, and dropout ratio at every GEN layer in the range of $\{0.1, 0.2, 0.3\}$. As a score function, we use DistMult [26] at the end of our GENs. For all datasets, we consider the inverse relation as suggested by several recent works for multi-relational graphs [11, 16, 20], since directed relation information flows along with both directions. Finally, to select the best model, we use the mean reciprocal rank (MRR) as an evaluation metric.

For **FB15k-237** dataset, we set the $\alpha = 1 \times 10^{-3}$ and $\gamma = 1$ with dropout rate 0.3. Also, we set the number of basis units $B = 100$ for the basis decomposition on each GEN layer, and sample 32 negative triplets for each positive triplet in both I-GEN and T-GEN. At every episodic training, we randomly sample 500 unseen entities in the meta-training set. Also, we validate and test models using all unseen entities in the meta-validation and meta-test sets, respectively.

Table 1: The naive and meta-learning strategy results of 1- and 3-shot OOG link prediction on FB15k-237 and NELL-995. Bold numbers denote the best results on I-GEN and T-GEN, respectively.

| Model | FB15k-237 | | | | | | | | NELL-995 | | | | | | | |
|---|---|---|---|---|---|---|---|---|---|---|---|---|---|---|---|---|
| | MRR | | H@1 | | H@3 | | H@10 | | MRR | | H@1 | | H@3 | | H@10 | |
| | 1-S | 3-S | 1-S | 3-S | 1-S | 3-S | 1-S | 3-S | 1-S | 3-S | 1-S | 3-S | 1-S | 3-S | 1-S | 3-S |
| I-GEN | **.348** | **.367** | **.270** | **.281** | **.382** | **.407** | **.504** | **.537** | **.278** | **.285** | **.206** | **.214** | **.313** | **.322** | **.416** | **.426** |
| w/o transfer strategy | .344 | .362 | .264 | .275 | .379 | .401 | .503 | .527 | .272 | .277 | .198 | .206 | .309 | .314 | .413 | .414 |
| T-GEN | **.367** | **.382** | **.282** | .289 | **.410** | **.430** | **.530** | **.565** | **.282** | **.291** | **.209** | **.217** | **.320** | **.333** | **.421** | **.433** |
| w/o transfer strategy | .362 | .381 | .278 | **.291** | .400 | .422 | .527 | .563 | .273 | .290 | .198 | **.217** | .310 | .326 | .412 | .431 |

For **NELL-995** dataset, we use the same settings with FB15k-237, except that we sample 64 negative triplets for each positive triplet.

For **WN18RR** dataset, we use the same settings with FB15k-237, except that we randomly sample 100 unseen entities for episodic training during meta-training.

**Drug-Drug Interaction** For both I-GEN and T-GEN, we search for the learning rate $\alpha$ in the range of $\left\{5 \times 10^{-4}, 1 \times 10^{-3}, 5 \times 10^{-3}\right\}$, and dropout ratio at every GEN layer in the range of $\{0.1, 0.2, 0.3\}$. As a score function, we use two linear layers with ReLU as an activation function at the end of the first layer. For all datasets, we consider the inverse relation as in the case of the knowledge graph completion task. Finally, to select the best model, we use the area under the receiver operating characteristic curve (ROC) as an evaluation metric.

For **DeepDDI** dataset, we set the $\alpha = 1 \times 10^{-3}$ with dropout rate 0.3 for both I-GEN and T-GEN. Also, we set the number of basis units $B = 200$ for the basis decomposition. At every episodic training, we randomly sample 80 unseen entities in the meta-training set. Also, we validate and test models using all unseen entities in the meta-validation and meta-test sets, respectively.

For **BIOSNAP-sub** dataset, we set the $\alpha = 1 \times 10^{-3}$ with dropout rate 0.1 for I-GEN and 0.2 for T-GEN, respectively. Also, we set the number of basis units $B = 200$ for the basis decomposition. At every episodic training, we randomly sample 50 unseen entities in the meta-training set. Also, we validate and test models using all unseen entities in the meta-validation and meta-test sets, respectively.

# B  Meta-learning for Long-tail Task

**Implementation Details** Many real-world graphs follow the long-tail distribution, where few entities have many links while the majority have few links (See Figure 1). For such an imbalanced graph, it would be beneficial to transfer the knowledge from entities with many links to entities with few links. To this end, we transfer the meta-knowledge on data-rich entities to data-poor entities by simulating the data-rich circumstance under the meta-learning framework, motivated by Wang et al. [22]. Specifically, we first meta-train our GENs with many shot cases (e.g. $K = 10$), and then gradually decrease the number of shots to few shots cases (e.g. $K = 1$ or 3) in logarithmic scale: $K_i = \lfloor \log_2(\text{max-iteration}/i) \rfloor + K$, where $K_i$ is the training shot size at the current iteration number $i$, and $K$ is the test shot size. In this way, GENs learn to represent the unseen entities using data-rich instances, and entities with few links regimes may experience like data-rich instances, with the model parameters trained on the entities with many links and tuned on the entities with few links.

**More Ablation Studies** Since knowledge graphs follow a highly long-tailed distribution (See Figure 1), we provide the more experimental results about transfer strategies on knowledge graph completion tasks, to demonstrate the effectiveness of the proposed meta-learning scheme on a long-tail task. Table 1 shows that the transfer strategy outperforms naive I-GEN and T-GEN on all evaluation metrics, except for only two H@1 cases of T-GEN on 3-shot OOG link prediction settings. We conjecture that the effectiveness of the meta-learning scheme is especially larger on 1-shot cases, where data is extremely poor, rather than the 3-shot cases.

Table 2: Total, seen-to-unseen and unseen-to-unseen results of 1- and 3-shot OOG link prediction on FB15k-237. * means training a model within our meta-learning framework. Bold numbers denote the best results.

| | | Total | | | | | | | | Seen to Unseen | | | | Unseen to Unseen | | | |
|---|---|---|---|---|---|---|---|---|---|---|---|---|---|---|---|---|---|
| | Model | MRR | | H@1 | | H@3 | | H@10 | | MRR | | H@10 | | MRR | | H@10 | |
| | | 1-S | 3-S | 1-S | 3-S | 1-S | 3-S | 1-S | 3-S | 1-S | 3-S | 1-S | 3-S | 1-S | 3-S | 1-S | 3-S |
| Seen to Seen | TransE [1] | .053 | .048 | .034 | .026 | .050 | .050 | .082 | .077 | .055 | .050 | .086 | .081 | .016 | .014 | .029 | .025 |
| | DistMult [26] | .017 | .014 | .010 | .009 | .019 | .014 | .029 | .022 | .018 | .015 | .029 | .022 | .011 | .007 | .025 | .015 |
| | R-GCN [16] | .008 | .006 | .004 | .003 | .007 | .005 | .011 | .010 | .003 | .003 | .005 | .006 | .076 | .050 | .101 | .070 |
| Seen to Unseen | MEAN [6] | .105 | .114 | .052 | .058 | .109 | .119 | .207 | .217 | .112 | .121 | .221 | .231 | .000 | .000 | .000 | .000 |
| | LAN [21] | .112 | .112 | .057 | .055 | .118 | .119 | .214 | .218 | .119 | .119 | .228 | .232 | .000 | .000 | .000 | .000 |
| | GMatching* [25] | .224 | .238 | .157 | .168 | .249 | .263 | .352 | .372 | .239 | .254 | .375 | .400 | .000 | .000 | .000 | .000 |
| Ours | I-GEN (Random) | .309 | .319 | .236 | 240 | .337 | .352 | .455 | .477 | .329 | .339 | .485 | .508 | .000 | .000 | .000 | .000 |
| | I-GEN (DistMult) | .348 | .367 | .270 | .281 | .382 | .407 | .504 | .537 | .371 | .391 | .537 | .571 | .000 | .000 | .000 | .000 |
| | I-GEN (TransE) | .345 | .371 | .259 | .275 | .385 | .416 | .515 | .559 | .367 | .395 | .548 | **.594** | .000 | .000 | .000 | .000 |
| | T-GEN (Random) | .349 | .360 | .268 | .273 | .385 | .398 | .508 | .532 | .361 | .373 | .529 | .554 | .168 | .164 | .185 | .192 |
| | T-GEN (DistMult) | **.367** | **.382** | **.282** | **.289** | **.410** | **.430** | .530 | **.565** | **.379** | **.396** | .550 | .588 | **.185** | **.175** | **.220** | .201 |
| | T-GEN (TransE) | .356 | .374 | .267 | .282 | .403 | .425 | **.531** | .552 | .368 | .387 | **.552** | .572 | .175 | **.175** | .205 | **.235** |

# C   Additional Experimental Results

**Effect of Score Function**   While we performed all experiments with DistMult score function in the main paper, we further evaluate proposed GENs on the few-shot OOG link prediction task with TransE [1], which is another popular score function. We use the same settings as with DistMult [26] experiments, except that we use TransE for the initial embedding and the score measurement. Table 2 shows that our I-GEN and T-GEN with TransE score function also outperform all baselines by impressive margins, where they perform comparably to DistMult. These results suggest that our model works regardless of the score function.

**Effect of Initialization**   We further demonstrate the meta-training effectiveness of our meta-learner, by randomly initializing an In-Graph, in which GEN extrapolates knowledge for an unseen entity without using the pre-trained embeddings of entity and relation. Table 2 shows that, while results with the random initialization are lower than pre-trained models, GENs are still powerful on the unseen entity, compared to the baselines. These results suggest that GENs trained under the meta-learning framework can be applied to more difficult situations, as pre-trained In-Graph might not be available for the few-shot OOG link prediction in real-world scenarios.

**Effect of Transductive Scheme**   As shown in Table 2, I-GEN achieves comparable performances with T-GEN on seen-to-unseen link prediction. However, since the inductive method can not handle two unseen entities at once, this scheme does not solve the unseen-to-unseen link prediction for two emerging entities. However, unseen entities do not emerge one by one, but may emerge simultaneously as a set in real-world settings, such that we consider a transductive scheme to deal with this challenging circumstance. Table 2 shows that, while the unseen-to-unseen link prediction performances of T-GEN is far from the seen-to-unseen performances, T-GEN can infer hidden relationships among unseen entities by transductive learning and inference.

Figure 2: T-SNE visualization of the learned embeddings for seen and unseen entities.

**More Visualization**   The experimental results on multiple datasets show that our GENs significantly outperform baselines, even when they are retrained with the unseen entities. To see why does GENs generalize well to the unseen entities, we visualize the output embeddings of seen-to-unseen baseline (LAN), seen-to-seen baseline (TransE) which is retrained from scratch, and T-GEN. As shown in

Table 3: Examples of the OOG link prediction on NELL-995. S: seen, U: unseen, O: correct prediction, X: incorrect prediction, (H): head entity, (R): relation, (T): tail entity, and ␣: unseen entity.

| Type | I-GEN | T-GEN | Triplet |
|------|-------|-------|---------|
| S-U | O | O | (H) musician_vivaldi,<br>(R) musician_plays_instrument,<br>(T) music_instrument_string |
| S-U | O | O | (H) city_hawthorne,<br>(R) city_located_in_state,<br>(T) state_or_province_california |
| S-U | O | O | (H) journalist_maureen_dowd,<br>(R) works_for,<br>(T) company_york_times |
| S-U | O | O | (H) person_monroe,<br>(R) person_born_in_location,<br>(T) county_york_city |
| S-U | O | O | (H) ceo_stan_o_neal,<br>(R) works_for,<br>(T) retailstore_merrill |
| S-U | O | O | (H) insect_insects,<br>(R) invertebrate_feed_on_food ,<br>(T) agricultural_product_wood |
| U-U | X | O | (H) person_katsuaki_watanabe,<br>(R) person_leads_organization,<br>(T) automobilemaker_toyota |
| U-U | X | O | (H) mlauthor_web_search,<br>(R) agent_competes_with_agent,<br>(T) website_altavista_com |
| U-U | X | O | (H) chemical_chromium,<br>(R) chemical_is_type_of_chemical,<br>(T) chemical_heavy_metals |
| U-U | X | X | (H) food_meals,<br>(R) food_decreases_the_risk_of_disease,<br>(T) disease_heart_disease |

Figure 2, since GEN embeds the unseen entities on the manifold of seen entities, it achieves better results on few-shot OOG link prediction tasks than baseline models.

## D   Examples

Table 3 shows some concrete examples of the OOG link prediction result from NELL-995 dataset, where the 7 to 9 rows show that our T-GEN correctly performs link prediction for two unseen entities.

## E   Discussion on Inductive and Transductive Schemes

In this section, we describe in detail about task-level transductive inference and meta-level inductive inference for the proposed transductive GEN (T-GEN) model. Since transductive GEN requires to predict links between two *unseen test entities* which is impossible to handle using conventional link prediction approaches, the problem is indeed transductive. Furthermore, the inference of unseen-to-unseen links could be also considered as inductive at meta-level, where we inductively learn the parameters of GEN across the batch of tasks. Thus, we are tackling transductive inference problems by considering them as meta-level inductive problems, but the intrinsic unseen-to-unseen link prediction is still transductive. To illustrate more concretely, different sets of unseen entities make mutually inconsistent predictions, which is caused by transduction. Other transductive meta-learning approaches such as TPN [9] and EGNN [7] tackle the problem with similar high-level ideas, where they classify unseen classes by leveraging both information on labeled and unlabeled nodes.