[Reviews · NeurIPS 2020]

Review 1

Summary and Contributions: Proposes a new model for few-shot out-of-graph link prediction. Experiments on two tasks are conducted to show effectiveness of the proposed model.

Strengths: The problem is important and interesting. The technical quality is overall good. The presentation is good.

Weaknesses: The novelty is incremental and the experiments should be improved.

Correctness: The method design is reasonable.

Clarity: Well written paper.

Relation to Prior Work: More related work should be compared and discussed.

Reproducibility: Yes

Additional Feedback: Detailed review -------------------- The manuscript proposes GEN, a graph extrapolation network for few-shot out-of-graph link prediction. GEN considers both relations between seen/unseen nodes and unseen/unseen nodes to learn embeddings, which is further optimized through VAE. Experiments on two different tasks (i.e., knowledge graph completion and drug-drug interaction) demonstrate that GEN outperforms some baseline methods in few-shot link prediction task. Pros 1. The problem is relatively new and important. It is meaningful to study few-shot link prediction problem in graphs. 2. The presentation is overall good. The content is clear for me. 3. A new model for few-shot link prediction is proposed. Two different tasks of different settings are conducted. Cons/Questions 1 The novelty of GEN is incremental. Using aggregators of both seen and unseen neighbors to learn node embeddings is not surprising. The main contribution lies in considering relations among unseen nodes. VAE for optimization is borrowed from previous work. No much novelty could be found. 2 Experiments should be improved. Baseline methods are relatively weak, it is better to compare and discuss more recent work for knowledge graph few-shot link prediction such as: Meta Relational Learning for Few-Shot Link Prediction in Knowledge Graphs, EMNLP 2019 Neural Snowball for Few-Shot Relation Learning, AAAI 2020 Few-Shot Knowledge Graph Completion, AAAI 2020 In addition, I think the datasets using in this work are relatively small (especially for drug-drug interactions with hundreds of nodes). It would be better if larger datasets could be used for evaluation. To summarize, the novelty of this work is incremental and the experiments could be improved. -------------------- After rebuttal -------------------- Add results of new baseline methods (MetaR and FSRL) in new version.


Review 2

Summary and Contributions: This paper presents a meta-learning framework for few-shot out-of-graph link prediction that tackles both inductive setting (learning links between a seen and an unseen entity) and transductive setting (learning links between two unseen entities). Their framework employs two GNNs: one to learn embeddings of unseen entities given a few examples, and one that has an additional stochastic layer on top integrate possible unseen-to-unseen relation links into the embeddings. They test their framework on two knowledge graph completion and two drug-to-drug interaction datasets, and demonstrate that their method does well in seen-to-unseen and unseen-to-unseen cases it aims to tackle.

Strengths: I think this is an interesting work on a difficult task, especially the unseen-to-unseen version of the evaluation. The experiments seem solid, especially given that they have evaluated their model on multiple datasets and two different domains. They also include analysis of the results and an ablation study.

Weaknesses: I think the biggest conceptual weakness is that the authors do not address the question: What is the advantage of using a meta-learning framework rather than retraining the entire model with the new triples? This is especially relevant given that some of the baseline models are extremely quick to train unless the dataset gets very large. A related question I would very much like to be answered experimentally is: how does this model compare to retraining the entire model from scratch? -- Update after author rebuttal -- I thank the authors for addressing this point and providing additional results comparing DistMult trained from scratch and their method. However, the reported numbers for the retraining baseline is quite low, and their meta-learning framework outperforms retraining significantly. This is very surprising to me, as I had expected the retraining to do better at the expense of training time. It would be great if the authors would include the details of the experimental setup, and some insight into this discrepancy.

Correctness: The method is presented with its relevant mathematical derivations and seems sound. Empirical methodology and evaluation also seems inline with standard practices, although I strongly recommend that the authors take a look at the recent paper "A Re-evaluation of Knowledge Graph Completion Methods", Sun et al. 2020 to make sure that their evaluation doesn't suffer from the problems outlined there.

Clarity: The paper is well written for the most part, however the technical parts are a little too dense in mathematical notation and not always easy to decipher. For example, the notation in the section on meta-learning framework would benefit from some simplification. I am confused about exactly how the initial embeddings from DistMult are incorporated into the model, can you make this clearer? I think the most interesting part of this paper is the stochastic inference for unseen-to-unseen entity links, however the section where this is presented is quite terse.

Relation to Prior Work: Previous relevant work is cited, and outlines the differences between the previous work and theirs.

Reproducibility: Yes

Additional Feedback: Minor comments: - General comment: make sure to introduce the full versions of acronyms before using them. - I am confused about the second sentence in intro (line 22): What does it mean for data to work in non-Euclidian domain? - Defn 3.1: term "relations" used to refer to both "r" and the triple (e_h, r, e_t) - line 134: I would argue that there is a subtle but important difference between maximizing the score of the true triples, and the standard KG task which is ranking, and aims to score the true triples above false ones. - line 153: There seems to be an implicit correspondence here between tasks and relations, is this correct? - line 173: typo: associative -> associated - line 203: noises -> noise - line 206: uncertainties -> uncertainty - line 295: You say your model trained on 1-shot setting obtains better performance on 3-shot setting, which seems quite surprising. Can you elaborate on this? - Table 3: "R-S"hard to understand. Which axis corresponds to training and testing also hard to understand, would be better if labelled.


Review 3

Summary and Contributions: This paper presents a novel neural network architecture aiming to extrapolate the learned model to unseen entities, involving seen-to-unseen and even unseen-to-unseen triplets. The proposed method transforms the original problem into a sequence of meta-learning tasks and then trains an inductive and a transductive network to represent the unseen entities with their neighbours (as a support set) in the set of seen entities.

Strengths: - This paper is motivated by an important problem in this area---extrapolation on unseen data, and the proposed framework appears sound; - The authors carried out experiments on two benchmark datasets, and the results are significantly better than the baseline approaches;

Weaknesses: - The proposed framework utilises the ideas of meta-learning and semi-supervised learning to solve the extrapolation problem. It uses the unseen entities' neighbouring entities and relations in the support set to generate the embedding of those unseen entities. This, however, relies on that the distribution of unseen data is still the same as the seen data, which might not be true in practical;

Correctness: The algorithm is reasonable, and results appears to be sound and correct.

Clarity: The paper is well written, and the algorithm is described clearly. However, there is something that should be improved: - Figure 1 is confusing, and the second subgraph is labelled as (d). Moreover, the left and centre subgraphs are talking about the same thing. The term "support set" and "query set" are not introduced until section 4. For the audience who are not familiar with semi-supervised learning, these terms only confuse them. Leaving Fig 1a in the introduction and move 1b to section 4 will be better for people to understand. - Equation 3 and 4 are inline instead of being displayed in another line, which is hard to find on paper; - I am surprised that the presented approach achieves much better results even on seen-to-seen link prediction comparing to the classic approaches like TransE. Does that mean the meta-semi-supervised-learning framework results in better embeddings? I would like to see more discussion on these results.

Relation to Prior Work: The paper has covered a broad range of related work, and made comparisons to many of them in the experiments.

Reproducibility: Yes

Additional Feedback:


Review 4

Summary and Contributions: 1. A new few-shot learning setting distinct from previous ones (e.g., Wiki-/NELL-one and FewRel), where entities scarcely appear on knowledge graph. 2. Based on meta-learning framework, the proposed method, namely GEN, takes into account both inductive and transductive scenarios.

Strengths: 1. A new few-shot learning setting with new datasets provides a way to evaluate meta-learning method, and the setting is related to real-world applications. 2. The evaluation results are good.

Weaknesses: 1. notations can be improved to be more accurate and simplified. For example, in line 163 eq1, here e_h and e_t represent an entity in general, however in line 161, the authors define e_h and e_t are seen entities. Too many notations and inconsistency can confuse the reader to some extent. And *simulated* unseen and *real* unseen sometimes are mixed up. An alternative way is to use meta-train and meta-test to denote the dataset during training and evaluation while during meta-training/meta-test the authors build support set and query set for every episode. This also raises my confusion in the meta-learning framework section: is the meta-training set includes unseen entities or not? For now I just assume it's not, otherwise, i believe the setting does not make any sense. Some notations miss explanation, such as \phi_i, f_{\theta} in eq2. Also, the figure1 includes too many lines and can be simplified for better readability. 2. how do you divide the support set and the query set in meta-training set? 3. Novelty of the model. I-GEN is pretty similar to meta-R and the T-GEN is a variant of graph-VAE. 4. What is Seen to Seen, …, Seen to Unseen in the most left col of table2? 5. It is better to denote clearly in the comparison that the T-GEN is under the transductive setting, which utilizes more data during evaluation. 6. why in Figure 5, more shots do not get improvements?

Correctness: Based on the authors’ proposed few-shot settings on several datasets, the proposed GEN gets state-of-the-art performance. However, no evaluation on benchmark dataset is reported on link prediction.

Clarity: Authors should revise this paper toward highlighting the most innovative part since through Section 4 GEN is only a combination of prior methods, e.g., MetaR, GraphVAE, from my perspective.

Relation to Prior Work: 1. Basically, metaR is applicable for the Few-shot setting proposed by authors, which can be employed as a strong baseline: [1] Meta Relational Learning for Few-Shot Link Prediction in Knowledge Graphs (MetaR) 2. More papers related (or with similar methodology) to this work: [2] Node Classification on Graphs with Few-Shot Novel Labels via Meta Transformed Network Embedding [3] Learning to Propagate for Graph Meta-learning [4] Graph Prototypical Networks for Few-shot Learning on Attributed Network

Reproducibility: Yes

Additional Feedback: The normal link prediction (FB15k-237 and WN18RR) datasets also include seen-to-unseen triples during testing. So, I’m wondering if the proposed method is able to improve the benchmark performance.

[Author Response · NeurIPS 2020]

We thank all reviewers for their constructive comments. We appreciate that reviewers find the proposed out-of-graph (OOG) link prediction problem to be important as well as novel [R1, R2, R3, R4], the paper well written [R1, R2, R3] and experimental results good [R2, R3, R4]. Due to the page limit, we address the major comments from the reviewers:

**Common Comments: Novelty over existing works on few-shot link prediction or GNNs [R1, R4].** We want to emphasize and clarify that our main contribution is neither proposing a general few-shot link prediction method nor a new GNN architecture for general purpose. As clearly stated in the introduction (Line 71-78), our contributions are as follows: 1) the proposal of the few-shot link prediction for **unseen entities** (seen-to-unseen, and unseen-to-unseen), 2) the **transductive meta-learning** framework to solve it by simulating the unseen entities with seen entities during meta-training. Thus any existing GNN models can be trained in our meta-learning framework (Table A).

**Experiments against more baselines [R1, R4].** [Chen et al. 19] and [Zhang et al. 20] tackle the prediction of unseen relations of **seen entities**, while our problem deals with **unseen entities**. Yet, we compared against the proposed baselines, MetaR [Chen et al. 19] and FSRL [Zhang et al. 20], on the 3-shot OOG link prediction task. The results in Table A show that they achieve extremely low performance compared to our GENs (Rows 2-3). Further, we trained the baselines in **our meta-learning framework** and obtained significantly improved results (Table A, Rows 4-6). However, their performances are still substantially lower than GENs, which show that GENs' dedicated embedding layers for seen-to-unseen and unseen-to-unseen link prediction are more effective for OOG link prediction. We will include Table A in the revision.

Table A: OOG link prediction results with more baselines. * denotes baselines trained with our meta-learning framework.

| | | FB15k-237 | | | | NELL-995 | | |
|---|---|---|---|---|---|---|---|---|
| | MRR | H@1 | H@3 | H@10 | MRR | H@1 | H@3 | H@10 |
| GMatching [51] | .105 | .061 | .112 | .183 | .079 | .059 | .097 | .106 |
| MetaR (Chen et al. 19) | .084 | .041 | .093 | .164 | .096 | .060 | .115 | .166 |
| FSRL (Zhang et al. 20) | .090 | .058 | .096 | .150 | .085 | .064 | .095 | .126 |
| GMatching* | .238 | .168 | .263 | .372 | .139 | .092 | .151 | .235 |
| Ours MetaR* | .316 | .235 | .341 | .492 | .213 | .145 | .247 | .352 |
| FSRL* | .259 | .186 | .281 | .404 | .161 | .106 | .181 | .275 |
| Ours I-GEN | **.367** | **.281** | **.407** | **.537** | **.285** | **.214** | **.322** | **.426** |
| T-GEN | **.382** | **.289** | **.430** | **.565** | **.291** | **.217** | **.333** | **.433** |

**Reviewer #1: Small datasets.** FB15k-237 and NELL-995 are large and contain 14,514 and 75,492 nodes respectively.

**Reviewer #2: Advantage of a meta-learning framework against retraining from scratch.** Our meta-learning framework enables to embed unseen entities without additional re-training which is efficient, and generalizes well to unseen entities. We additionally compared GENs against models trained from scratch (Table B, top two rows), which GENs largely outperform with a fraction of time required to embed unseen entities (Table B). MetaR trained in our meta-learning framework is slower since it uses additional gradient information.

Table B: Retraining.

| | MRR | Time |
|---|---|---|
| DistMult | .094 | 158.25 sec |
| TransE | .120 | 185.09 sec |
| MetaR* | .316 | 13.97 sec |
| I-GEN | .367 | **0.99 sec** |
| T-GEN | **.382** | 1.12 sec |

**Evaluation by the protocol in Sun et al. 20.** We found that the performance of our model remains consistent across both Top and Bottom evaluation protocols proposed in Sun et al. 20, and exactly the same as the reported performance.

**DistMult initialization.** We initialize the pre-trained embedding of seen entities and relations from DistMult for efficient training. However, we also report the results with random initialization in Table 2 of the supplementary file.

**Reviewer #3: The distribution of unseen data might not be the same as the seen data.** While the initial distribution of unseen data might not be the same as the seen data, when we want to infer relationships among entities, unseen entities should be closer to embeddings of seen entities. When looking at the performance results in Table 2. and the embedding results in Figure A, it can be seen that the performance is good when the unseen data is well aligned with the seen data.

**GENs result in better embeddings over seen-to-seen training using TransE.** The visualization of the TransE embeddings (Figure A) shows that the embeddings for unseen entities trained in a seen-to-seen manner are not aligned with seen entities, while GEN aligns the unseen entities with the seen entities.

Figure A: T-SNE visualization.

**Reviewer #4: Does the meta-training set includes unseen entities?** The meta-training set **does not** include real unseen entities at meta-test time, and we simulate the unseen entities with a subset of seen entities during meta-training. We will use the term "simulated unseen" and "real unseen" for further clarification.

**How to divide the support set and the query set.** We randomly sample the K-triplets associated with each entity for a support set at every episode, and the remaining samples are used as a query set (Line 172-177) in meta-training.

**Seen to Seen and Seen to Unseen in Table2?** They denote the baseline types. The seen-to-seen are baselines that only handle seen entities, and seen-to-unseen baselines are ones that can tackle seen-to-unseen link prediction tasks.

**Denote that the T-GEN utilizes more data during evaluation.** T-GEN does not utilize more data, since it performs unseen-to-unseen link prediction for exactly the same set of entities given to all methods. We will clarify this.

**No improvements with larger shots (Figure 5).** This behavior is consistent with the baselines, and is due to larger shots introducing more noise from weakly-related neighboring nodes.

**Can the proposed method improve the benchmark performance?** They do, but the ratio of the seen-to-unseen triplets is very small. For example, WN18RR dataset has only 16 seen-to-unseen triplets (0.02%) to evaluate on a query set. Thus, we compared GENs only against seen-to-unseen triplets for WN18RR dataset (Table C). The results demonstrate that the seen-to-unseen performance of the benchmark datasets can also be improved using our GEN.

Table C: Results on WN18RR.

| | MRR | H@1 |
|---|---|---|
| DistMult | .000 | .000 |
| TransE | .011 | .000 |
| MetaR* | .066 | .063 |
| I-GEN | **.125** | **.125** |

[Meta-Review · NeurIPS 2020]

This paper consider few-shot link prediction in the "out of graph" setting, in which novel nodes that were not available at training time are introduced. The reviewers initially had questions about the novelty of the approach and the baselines considered in the experiments. These questions were sufficiently addressed in the authors' response, which clarified that the emphasis on the out of graph setting was the main novelty and provided additional experiments to show that other baselines did not handle this problem as well. The reviewers strongly encourage the reviewers to add these results and clarifications to the camera ready version of the paper. After discussion, they agreed that the paper was ready for publication and made a novel contribution to an interesting problem that has not received much attention. We note that Reviewer 2 asks some follow up questions about the new results and the authors are also strongly encouraged to address them.